# The Basics and the Advancements in Diagnosis of Bacterial Lower Respiratory Tract Infections

**DOI:** 10.3390/diagnostics9020037

**Published:** 2019-04-03

**Authors:** Stephanie Noviello, David B. Huang

**Affiliations:** 1Motif BioSciences, Princeton, NJ 08540, USA; david.huang@motifbio.com; 2Department of Internal Medicine, Division of Infectious Diseases, Rutgers New Jersey Medical School, Trenton, NJ 07103, USA

**Keywords:** rapid diagnostic, bacterial infection, lower respiratory tract infection

## Abstract

Lower respiratory tract infections (LRTIs) are the leading infectious cause of death and the sixth-leading cause of death overall worldwide. *Streptococcus pneumoniae*, with more than 90 serotypes, remains the most common identified cause of community-acquired acute bacterial pneumonia. Antibiotics treat LRTIs with a bacterial etiology. With the potential for antibiotic-resistant bacteria, defining the etiology of the LRTI is imperative for appropriate patient treatment. C-reactive protein and procalcitonin are point-of-care tests that may differentiate bacterial versus viral etiologies of LRTIs. Major advancements are currently advancing the ability to make rapid diagnoses and identification of the bacterial etiology of LRTIs, which will continue to support antimicrobial stewardship, and is the focus of this review.

## 1. Who is at Risk of Lower Respiratory Tract Infections (LRTIs) and Their Outcomes?

In 2016, there were an estimated 336 million cases of lower respiratory tract infections (LRTIs) globally [1]. LRTIs are the leading infectious cause of death and the sixth-leading cause of death overall worldwide [1]. In 2016 alone, LRTIs (defined as pneumonia, bronchitis, or bronchiolitis) caused an estimated 2.38 million deaths with a disproportionate effect on children younger than 5 years of age and adults >70 years old [1]. Pneumonia due to *Streptococcus pneumoniae* caused over half of these deaths in all ages, approximately 1.2 million deaths [1]. Globally, improvements in mortality rates have been seen among children <5 years old, but this is offset by increases in LRTI disease burden among people >70 years old. Typically, bacterial infections are associated with higher mortality rates from LRTIs compared to other infections [1].

## 2. What Causes Bacterial Lower Respiratory Tract Infections?

LRTIs include the diagnoses of pneumonia, bronchitis, or bronchiolitis (in young children) and other lung alveolar and airway infections, and are caused by bacteria, viruses, fungi and even parasites. In the pre-antibiotic years, the bacterial etiology for pneumonia was most commonly (> 90%) due to *S. pneumoniae*, but the incidence of bacterial etiologies for pneumonia has been decreasing in recent years especially with the availability of vaccines like the pneumococcal polysaccharide vaccine (PPSV23 or Pneumovax^®^) and the pneumococcal conjugate vaccine (PCV13 or Prevnar 13^®^) [2]. Currently <15% of cases of community-acquired pneumonia are due to *S. pneumoniae*, likely due to increases in vaccination rates against this pathogen [3]. Viral etiologies are now identified in 25% of patients, with many of these patients having bacterial coinfections with viral respiratory pathogens resulting in decreased mucociliary clearance of bacteria [3,4]. Examples of this pathology include influenza decreasing clearance of *S. pneumoniae*, and respiratory syncytial virus (RSV) inducing the adherence of *S. pneumoniae*, *Pseudomonas aeruginosa*, and *Haemophilus influenzae* to airway epithelial cells [4]. Complicating the diagnoses of patients, some respiratory pathogens are frequently detected in healthy patients with colonization, resulting in a questionable causative role in LRTIs [5].

Community-acquired pneumonia (CAP) is a major respiratory disease with a high prevalence in the general population. The Centers for Disease Control and Prevention in the United States evaluated the etiology of pneumonia in the community (EPIC study) between 2010 and 2012. The financial burden of pneumonia in the United States was estimated to be approximately $10 billion [6]. A pathogen was identified for 38% (853/2259) of adult patients who had radiologic evidence of CAP and an available specimen for diagnosis. For the 853 patients in whom a pathogen was identified, the etiologies included viral (62%), bacterial (29%), both viral and bacterial (7%), and fungal or mycobacterial (2%) pathogens with the most common being rhinovirus, influenza, and *S. pneumoniae* (Figure 1) [6]. *S. pneumoniae*, with more than 90 capsular serotypes, remains the most commonly identified cause of community-acquired acute bacterial pneumonia [7]. In the European Union (EU), about 3,370,000 cases of CAP are expected annually with hospitalization rates ranging between 20% and 50% [8]. Other bacterial etiologies of pneumonia include *Streptococcus pyogenes*, *Staphylococcus aureus*, *Haemophilus influenzae* (usually nontypable) and *Klebsiella pneumoniae,* although these are uncommon and often associated with a comorbidity, such as chronic obstructive lung disease or alcoholism. In a 3-year prospective study conducted in Finland, the etiology of CAP in children was detected in 86% of 254 children with 62% being viral, 53% bacterial, and 30% coinfection [9]. *S. pneumoniae* was the most commonly identified causative pathogen [9].

The incidence of hospital-associated pneumonia (HAP) is approximately 0.5–2.0% of hospitalized patients in the United States. While it is the second most common nosocomial infection, accounting for 22% of all hospital-associated infections, it is the deadliest with case-fatality rates between 30% and 70% [10]. Approximately 20% (23/110) of nosocomial pneumonia infections do not have a pathogen reported. Among those with a pathogen reported, the most common cause of nosocomial pneumonia is *S. aureus* (Figure 1). Even in HAP, generally considered to be less severe than ventilator-associated pneumonia (VAP), serious complications occur in approximately 50% of patients, including respiratory failure, pleural effusions, septic shock, renal failure, and empyema [11]. Based on data reported to the National Healthcare Safety Network at the Centers for Disease Control and Prevention, 2011–2014, *S. aureus* accounts for 25% of VAP, making it the most common cause of VAP [12]. Other pathogens which lead to HAP/VAP include *Pseudomonas aeruginosa*, enteric Gram-negative bacilli, and *Acinetobacter baumannii* [13].

Identifying causative pathogens in acute bronchitis is also quite difficult, with <30% of cases having a causative pathogen [14]. Up to 10% of acute bronchitis is due to bacteria, including *Bordetella pertussis*, *Chlamydia pneumoniae*, and *Mycoplasma pneumoniae*, whereas approximately 90% of cases are due to viral infections such as adenovirus, coronavirus, parainfluenza, influenza, and rhinovirus [15]. Bronchiolitis which occurs for the large part in infants is due to viral etiologies with the most common being RSV; most children have had an RSV infection by 2 years old [16].

## 3. How to Differentiate Bacterial from Viral LRTI at the Point of Care?

Antibiotics treat LRTIs with a bacterial etiology. With the potential for antibiotic-resistant bacteria as well as sequelae from the use of unnecessary antibiotics including *Clostridioides difficile* infection, defining the etiology of the LRTI is imperative for appropriate patient treatment. Currently, there are few diagnostic tools to adequately do this in a time-efficient manner at the point of care.

Clinical assessment does not typically decipher between bacterial, viral or both as an etiology for LRTIs. Therefore, diagnostic tools are essential for empiric treatment. Currently, these tools include the use of C-reactive protein, procalcitonin, and/or other combinations.

Briefly, C-reactive protein (CRP) is an acute phase reactant synthesized by the liver in response to cytokines, such as interleukin-6, released by macrophages and adipocytes in response to inflammatory conditions from bacterial infections. Consortia have developed interpretative cut-offs for CRP levels to assist physicians with antibiotic prescribing. CRP levels ≤ 20 mg/L indicate a self-limited LRTI for which antibiotics are not needed, and CRP ≥ 100 mg/L indicate severe infection for which antibiotics should be prescribed [17,18]. CRP levels between 21 and 99 mg/L are more challenging to interpret and must include further clinical assessment (Table 1).

Although rapid tests for CRP are used in point-of-care settings, the use of CRP has been controversial. A Cochrane review of trials conducted throughout Europe and Russia determined that CRP levels may reduce the use of antibiotics but the results did not affect patient outcomes, and suggested that increased hospitalization due to CRP evaluation may occur [20]. Although Andreeva et al. reports a decrease of 36% in antibiotic prescribing with the evaluation of CRP, the authors discuss multiple studies that have not resulted in such changes [21]. Therefore, the utility of CRP levels remains specific to individual treatment settings, and the measurement of CRP is not a substitute for clinical assessment and follow-up, which remain main-stays in the assessment of LRTIs.

For HAP/VAP, Infectious Diseases Society of America (IDSA) has indicated that clinical criteria alone, rather than using CRP is preferred, since CRP results did not reproducibly determine whether VAP was bacterial, leaving clinicians to rely on clinical assessment alone [13]. Procalcitonin (PCT) is another acute phase reactant associated with bacterial infections. PCT increases within 2–4 h of infection, peaking at 24–48 h. PCT is used to assist in the diagnosis of sepsis and has since been used for LRTIs and post-operative infections. Like CRP, its use has been targeted to ensure appropriate antibiotic use (Table 1). Typically, PCT is produced by parafollicular cells of the thyroid and by the neuroendocrine cells of the lung and the intestine in small quantities and is a precursor to calcitonin which regulates calcium and phosphate in the blood, but bacterial endokines and cytotoxins stimulate its production early in the disease process. Evidence has shown that PCT is a useful method in guiding the initiation and duration of antibiotic treatment for LRTIs [22]. A meta-analysis of 32 randomized studies with a majority of patients with acute LRTIs showed that PCT testing lowered mortality (decrease of 1.4%), antibiotic consumption (2.4 day mean reduction in exposure), and antibiotic-related adverse events (decrease of 5.8%) [23]. Briel et al. evaluated 458 patients whom the physician thought needed antibiotics for a respiratory tract infection [24]. Patients were randomized to PCT-guided approach to antibiotic therapy or to a standard approach. The antibiotic prescription rate was 72% lower in those who had procalcitonin-guided antibiotic use without any impact on patient outcome. However, Huang et al. conducted a study in 14 hospitals in the United States and among 1656 patients observed no significant difference between the PCT group and the usual-care group in antibiotic days (mean, 4.2 and 4.3 days, respectively) or the proportion of patients with adverse outcomes (11.7% and 13.1%, respectively) [19]. The bioMérieux’s VIDAS^®^ BRAHMS PCT™ test has been developed and was approved by FDA in 2017 to differentiate bacterial from viral infections and ultimately whether antibiotics are needed for pneumonia (Table 1) [25]. An ongoing study (Targeted Reduction of Antibiotics using Procalcitonin; TRAP-LRTI) is evaluating outpatient adults with suspected LRTIs and low procalcitonin levels [26]. Low blood levels of PCT (≤0.25 ng/mL) using bioMérieux’s VIDAS^®^ BRAHMS PCT™ test, which produces results within 20 min, is being used as an inclusion criterion, and then patients will be randomized to either azithromycin for 5 days or placebo. At Day 5, patients will be evaluated for improvement in symptoms with additional follow-up to 28 days after randomization. The study will evaluate the recovery of patients given azithromycin versus placebo, and whether a low PCT level can be used to avoid antibiotic therapy. The study will be completed in 2020, and it will add evidence to the utility of point-of-care PCT testing for patients with symptoms of LRTI in the outpatient setting [27].

Using host biomarkers in conjunction has also been studied and found to have high sensitivity and specificity for bacterial LRTIs [28,29]. A point-of-care test of CRP and Myxovirus resistance protein A (MxA) was used in 54 patients with pharyngitis or LRTIs to determine the etiology of the infection [29]. This combination characterized 80% (16/20) with bacterial infection, 70% (7/10) with viral infection, along with 92% (22/24) negative for a bacterial or viral infection. However, this study was small, and further confirmation of this point of care test is needed. Another host-protein signature assay combines the results of tumor necrosis-factor related apoptosis-inducing ligand (TRAIL), interleukin-10, and CRP and produces a score of 0–100 using the ImmunoXpert™ software. ImmunoXpert™ scores of <35 indicate nonbacterial etiology, whereas scores of ≥65 predict bacterial infections including mixed viral/bacterial co-infections [30]. This assay has a sensitivity of 93% with a 91–94% specificity. The use of this assay was superior to using the biomarkers individually, so development is continuing for a point-of-care platform to provide results within 15 min [30].

## 4. How to Determine the Bacterial Pathogen of LRTIs?

Major advancements in the diagnosis of bacterial LRTIs have occurred over the past ten years with the field still evolving. Etiologic determination of pneumonia and other LRTIs is typically challenging based on clinical assessment alone [31]. In addition, the collection of optimal specimens to detect the pathogenic etiology of LRTIs must be considered, given the specifics of a testing modality as well as the logistics of obtaining the specimens. Specimens can be collected using invasive (blood, thoracentesis, transthoracic needle aspiration, bronchoscopic bronchoalveolar lavage, or protected specimen brush) or noninvasive techniques (induced or expectorated sputum, nasopharyngeal swab, oropharyngeal/throat swab, and urine for antigen testing) [32]. However, colonization of the respiratory tract with various pathogens must be taken into account when determining appropriate treatment regimens based on sputum or naso/oropharyngeal swabs. In a study of 340 patients with CAP, culture and RT-PCR were used to compare plasma and respiratory (nasopharyngeal swabs and sputum or tracheal aspirates) samples [33]. In this study, RT-PCR for both plasma and respiratory samples identified *S. pneumoniae* more often than culture. However, good quality sputum samples are often challenging to obtain especially in the outpatient setting and from children.

The gold standard for identification of bacterial, viral, and fungal pathogens remains culture. For bacterial pathogens, testing for antimicrobial susceptibility should also be conducted to ensure adequate therapy is being administered.

However, culture and susceptibilities often take multiple days to obtain results; days which can include exposing patients to possibly ineffective therapies with significant safety repercussions. Initial inappropriate treatment has been identified as a risk factor which increases mortality rates in patients with HABP/VABP [34]. Therefore, other diagnostic methods have been evaluated and assist in providing timely diagnoses (Table 2).

Matrix-assisted laser desorption/ionization time-of-flight mass spectrometry (MALDI-TOF MS) is another modality used to diagnose bacterial LRTIs [52]. Approximately one isolated single colony from a culture plate is analyzed by the MALDI-TOF MS automated workflow and results are available within a few minutes identifying the microorganism based on matching to a library of microorganisms. Two MALDI-TOF MS systems have been approved by Food and Drug Administration (FDA) and include the Vitek MS from bioMérieux, Inc, which can identify 1046 species including mycobacteria, as well as the MALDI Biotyper CA System from Bruker Daltonics, Inc, which can identify 333 species or species groups representing over 424 bacteria and yeast species. The species identified are different between the systems [35,36]. In a study comparing the two systems, both systems correctly identified over 85% of the strains tested (254 Gram-positive bacteria, 167 Gram-negative bacteria, 109 mycobacteria and aerobic actinomycetes and 112 yeasts and mold), which included microorganisms associated with LRTIs [53]. The MALDI Biotyper CA is a smaller desktop technology compared with the larger Vitek MS. The drawback of the MALDI-TOF MS technology is the potential for overdiagnosis, depending on the colonies that are selected for testing, and lack of ability to identify new pathogens.

Primary antibiotic susceptibility testing is conducted using either automated or Kirby–Bauer tests. ETEST^®^ is a method to supplement testing with antimicrobial susceptibility testing (AST) reagent strips that determine minimum inhibitory concentrations [54]. Recently, SeLux has developed a Next Generation Phenotyping (NGP) platform, which is a high-throughput, fully-automated AST testing system, enabling same-shift susceptibility testing of up to 50 antibiotics in parallel [55]. This assay differentiates antibiotic-induced bacterial growth modes with a surface-binding fluorescent amplifier.

Serological tests have not been particularly sensitive nor specific when it comes to diagnosing atypical bacteria, such as *M. pneumoniae* [56]. However, nucleic acid amplification tests (NAATs) are another testing modality which include standard polymerase chain reaction (PCR) that provide rapid, highly sensitive and specific results. Recent additions to the NAATs’ diagnostic armamentarium include PCR with multiplex and real-time readings which have been FDA-approved for the diagnosis of LRTIs. These panels test a sample and obtain results within <5 h, reducing the time needed to confirm a causal pathogen for LRTIs. The Unyvero LRT cartridge, which was approved in 2018 by FDA, detects pathogens associated with >90% of pneumonia in hospitalized patients as well as genetic antibiotic resistance markers in endotracheal aspirate samples [57]. The LRT cartridge panel has a sensitivity of 91.4% and a specificity of 99.5% across all lower respiratory tract panel pathogens. It is a first-in-class molecular test for LRTIs and is the first automated molecular diagnostic test approved by FDA for *Legionella pneumoniae*. Future FDA approval is being sought for expanding the samples to include bronchoalveolar lavage aspirates and expand the assay to include *Pneumocystis jirovecii*. In addition, development of a smaller unit, Unyvero A30 RQ is in process with faster results (within 90 min) using real-time PCR.

Another real-time PCR test, which was recently approved by the FDA in 2018 includes the BIOFIRE FilmArray System (RT-PCR/nested multiplex PCR) Pneumonia Panel plus by bio Mérieux [38]. This assay can detect 18 bacteria (11 Gram negative, four Gram positive, and three atypical), seven antibiotic resistance markers, and nine viruses that cause pneumonia and other LRTIs and seven genetic markers of antibiotic resistance within 1 h, using sputum (including endotracheal aspirate) and bronchoalveolar lavage (including mini-BAL) sample types. The sensitivity of the assay is 96% with a specificity of 97%. The Pneumonia Panel received FDA clearance and CE-Marking in November 2018.

Although PCR testing can be compromised due to contamination, the quantitation of results is helpful in determining whether the result is due to contamination or is the clinically relevant infectious etiology. Some of these approved panels provide semi-quantitative results, such as the BIOFIRE FilmArray System Pneumonia Panel plus, which reports levels of organism concentration in genome copies/mL for values above 10^3.5^ copies/mL for 15 bacterial pathogens [38]. In addition, some NAATs can provide both viral and bacterial rapid assessments, which is particularly helpful in diagnosing LRTIs, namely pneumonia with infiltrate(s) on chest x-ray. Bacterial and viral coinfection certainly need to be ruled-out in patients with pneumonia. Therefore, this ability to determine both viral and bacterial etiologies is advantageous.

However, NAATs do have limitations, including the constant need to keep its internal references updated for bacteria and viruses. Metagenomic sequencing-based shotgun diagnostics do not compare the sample to known organisms or resistance patterns in internal databases. Rather, these shotgun metagenomics methods involve extraction of total DNA and/or RNA (usually followed by conversion to DNA) from primary specimens, fragmentation, library preparation, and depth sequencing [58]. Charalampous, et al. report on an optimized nanopore sequencing-based clinical metagenomics test (Oxford Nanopore Technologies), which removed approximately 99.99% of host DNA from clinical respiratory samples (a challenging specimen given the low pathogen load compared with the backdrop of host nucleic acids (up to 1:10^5^ in sputum)), enabling pathogen genome assemblies equivalent to whole genome sequencing of isolates within 48 h [59]. This would allow for identification of new pathogen emergence, unusual resistance patterns, as well as outbreak assessment. This method of testing currently remains in development.

Urine antigen testing is also available for select pathogens, and is rapid with results in less than 1 h. Enzyme immunoassay and lateral flow assays are used to detect serogroup 1 of *Legionella pneumophilia*, the strain most common to cause infection, and enzyme immunoassay is available for *S. pneumoniae* detection [46,47,48,49,50].

## 5. Why are Diagnostics Important for LRTIs? (Antibiotic Stewardship)

Knowing which patients with LRTIs to treat and not to treat is challenging to determine, and physicians often err on the side of caution and prescribe antibiotics, given the high mortality rates of some bacterial LRTIs often without diagnostic results. Most patients are then empirically treated with antibiotics to pre-emptively avoid severe complications from bacterial LRTIs.

Improved diagnosis of the etiology of these infections would enable targeted therapy, leading to an overall more judicious use of antibiotics, which would likely decrease the rate of antimicrobial drug resistance as well as the safety impact of inappropriate treatment modalities on the patient [60]. Due to the improper treatment of LRTIs, some infected patients may not be treated adequately because the responsible bacterium (such as *S. pneumoniae,* methicillin-resistant *S. aureus* and Gram-negative bacilli) is resistant to available antibiotics, leaving physicians without a weapon to combat the illness [61,62]. The prudent use of available antibiotics in patients and animals, giving them only when needed, with the correct diagnosis and etiologic understanding, and in the correct dosage, dose intervals and duration is imperative. Antimicrobial stewardship is based on this premise. Over 262 million courses of outpatient antibiotic therapy were prescribed in 2011 with half of those antibiotics being unnecessary [63]. The most inappropriate use is for acute respiratory infections, including acute bronchitis. Further research into rapid, patient-friendly, inexpensive, and accessible diagnostic modalities to appropriately characterize LRTIs as bacterial versus viral versus other is necessary to harness antibiotic use. In addition, determination of the causative bacterial pathogen will further antibiotic stewardship programs, lowering the risk of propagating resistance and unwanted adverse events including the development of *C. difficile*. The advancements noted above are certainly moving in the right direction to understanding the etiology of pneumonia in a rapid manner, but development still continues for even faster, more comprehensive testing.

## Figures and Tables

**Figure 1 diagnostics-09-00037-f001:**
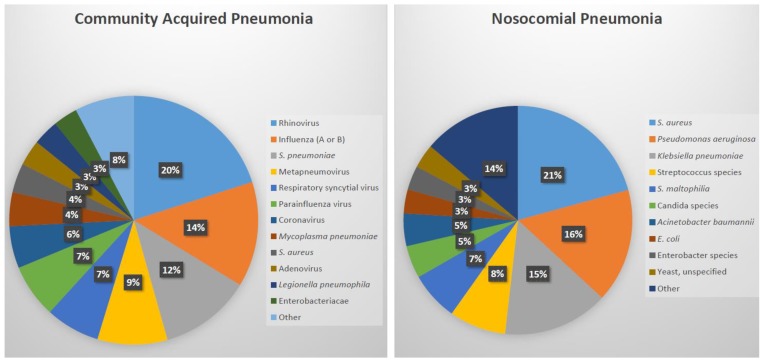
Identified pathogens in community acquired pneumonia (n = 966 pathogens in 853 adults) and nosocomial pneumonia (n = 87) [6,10].

**Table 1 diagnostics-09-00037-t001:** C-reactive protein and procalcitonin levels and need for antibiotic therapy [18,19].

C-Reactive Protein Value (mg/L)	Antibiotic Therapy
<20	Withhold in most patients
21–99	Further assessment needed to determine
≥100	Strongly encouraged
**Procalcitonin Value (ng/mL)**	
< 0.10 ng/mL	Strongly discouraged
0.10–0.25 ng/mL	Discouraged
0.26–0.50 ng/mL	Encouraged
> 0.50 ng/mL	Strongly encouraged

**Table 2 diagnostics-09-00037-t002:** Select diagnostics for bacterial lower respiratory tract infections (LRTIs).

Diagnostic	Pathogens	Company	FDA Approved	Sample Type	Time
*Matrix-assisted laser desorption/ionisation time-of-flight mass spectrometry (MALDI-TOF MS)*					
MALDI Biotyper CA; microflex LT/SH MALDI-MS; IVD MALDI Biotyper; MALDI Biotyper smart System; MBT smart CA System [35]	Bacterial, Fungal; 333 species or species groups	Brukner Daltonics	Yes	Induced or expectorated sputum, nasal aspirates or washes, nasopharyngeal (NP) swabs or aspirates, throat washes or swabs, bronchoscopic specimens	Within minutes of analyzing a single colony from isolate
Vitek MS [36]	Bacterial, Fungal; 1316 species, includes *Brucella, Candida auris, Elizabethkingia anophelis*	bio Mérieux	Yes	Induced or expectorated sputum, nasal aspirates or washes, NP swabs or aspirates, throat washes or swabs, bronchoscopic specimens	Within minutes of analyzing a single colony from isolate
*Nucleic acid amplification tests - Polymerase chain reaction*					
Unyvero A50 System [37]	>30 Gram-positive/Gram-negative bacteria and 10 antibiotic resistance markers or toxins	Curetis AG	Yes	Induced or expectorated sputum, nasal aspirates or washes, NP swabs or aspirates, throat washes or swabs, bronchoscopic specimens	<5 h
BIOFIRE FilmArray System Pneumonia Panel plus [38]	11 Gram-negative, 4 Gram-positive and 3 atypical bacteria, 9 viruses, 7 genetic markers of antibiotic resistance	bio Mérieux	Yes	Induced or expectorated sputum; endotracheal aspirates, bronchoscopic specimens	1 h
Multiplex one-step RT-PCR - FTD Respiratory pathogens 33 [39]	22 viruses, 11 bacterial pathogens	Fast Track Diagnostics	Yes	Induced or expectorated sputum; endotracheal aspirates, bronchoscopic specimens	<2 h
Multiplex ligation-dependent probe amplification - RespiFinder 22 [40]	18 viruses, 4 bacterial pathogens	PathoFinder	Yes	Induced or expectorated sputum; endotracheal aspirates, bronchoscopic specimens	<2 h
VERIGENE Respiratory Pathogens *Flex* Test [41]	13 viral and 3 bacterial (*Bordetella sp*) targets	Nanosphere/Luminex Corporation	Yes	Induced or expectorated sputum; endotracheal aspirates, bronchoscopic specimens	<2 h
*Serological tests*					
*Enzyme immunoassay*	*Mycoplasma pneumoniae* IgG and IgM	Vircell [42]/Zeus [43,44]	Yes	5 mL serum	<2 h
*Microimmunofluorescent stain*	*Chlaymydia pneumoniae*	MRL Diagnostics [45]/Labsystems [46]	Yes	5 mL serum	<2 h
*Urine antigen tests*					
Enzyme immunoassay - BinaxNOW *Legionella* Urinary Antigen Card [47]	*Legionella pneumophilia* (for serogroup 1)	Alere/Abbott	Yes	10 mL of urine	<1 h
Lateral Flow Assay - SAS *Legionella* [48]	*Legionella pneumophilia* (for serogroup 1)	SA Scientific	Yes	10 mL of urine	<1 h
Lateral Flow Assay - Bartels *Legionella urinary antigen* [49]	*Legionella pneumophilia* (for serogroup 1)	Trinity Biotech	Yes	10 mL of urine	<1 h
Lateral Flow Assay - Meridian Tru *Legionella* Assay [50]	*Legionella pneumophilia* (for serogroup 1)	Meridian BioSciences	Yes	10 mL of urine	<1 h
Enzyme immunoassay – BinaxNow *Streptococcus pneumoniae* Antigen Card [51]	*S. pneumoniae*	Alere/Abbott	Yes	10 mL of urine	<1 h

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
