# Peer review of "The Basics and the Advancements in Diagnosis of Bacterial Lower Respiratory Tract Infections"

_diagnostics, 2019, doi:10.3390/diagnostics9020037_

Reviewer 1 Report

page 1, line 31: instead of »(in children)« it should be »(in young children)«

per definition, bronchiolitis occurs in children<24 months of age, but recent trials have enroled only children <12 months of age

page 1, line 37: please provide a reference for the frequency of S. pneumoniae as a causative agent of pneumonia

page 1, line 40: I would suggest adding a sentence explaining that since certain respiratory viruses are frequently detected in healthy patients, their causative role in LRTI has been questioned (reference: Self et al. J Infect Dis. 2016; 213; 584-91. doi: 10.1093/infdis/jiv323)

page 2, line 49: I believe there are more than 90 pneumococcal serotypes

page 2, line 53: should be Klebsiella pneumoniae

page 2, line 54: would suggest adding a sentence related to etiology of CAP in children, where a pathogen was detected in 86% of 254 children with CAP in a relatively old but a well performed prospective study from Finland (Juven T et al. Pediatr Infect Dis J 2000; 19:293-8.

page 2, line 57: mortality should be changed to case-fatality rate

page 2, Figure 1, left image: instead of Pseudomonas aeruginosa it should be aeruginosa

page 3, line 100: maybe include a sentence (and a reference) about temporal evolution of CRP during LRTI/pneumonia, i.e. it being low in the first 12-24 hours and reaching peak values on day 2-3 of illness.

page 4, line 136: there is another POCT using MxA and CRP that has been evaluated in patients with URTI and LRTI (Eur Clin Respir J. 2015; 2: 10.3402/ecrj.v2.28245.); moreover, RNA signatures have also been evaluated in children

page 4, line 141: what is a relevant sample for determining etiology of LRTI? blood culture is relevant, BAL as well, tracheal aspirate is not always representative, sputum quality depends on patient effort and is useless in children. There have been studies looking for S. pneumoniae in blood with the use of PCR and it was superior to blood culture (Cvitkovič-Špik et al. Scand J Infect Dis 2013;45:731-7). I think these questions deserve some comments and discussion (with references) because they are clinically relevant.  There was a study using transthoracic aspirates that determined etiology of CAP in children in 47% of culture-negative patients (Vuori-Holopainen et al. Clin Infect Dis 2002; 34: 583-90.) Although the procedure seemed dangerous it was not associated with relevant complications.

page 5, Table: type of sample should be included with every test (in a separate column)

page 7, line 209: urine tests are not mentioned, although they are both reliable and fast, especially in adult patients with CAP: urine pneumococcal antigen and Legionella antigen – these tests require mentioning since they give reliable results rapidly and provide important information regarding selection of antibiotics

Author Response

REVIEWER 1

page 1, line 31: instead of »(in children)« it should be »(in young children)«

per definition, bronchiolitis occurs in children<24 months of age, but recent trials have enroled only children <12 months of age

RESPONSE: Text updated.

page 1, line 37: please provide a reference for the frequency of S. pneumoniae as a causative agent of pneumonia

RESPONSE: Reference added. 

page 1, line 40: I would suggest adding a sentence explaining that since certain respiratory viruses are frequently detected in healthy patients, their causative role in LRTI has been questioned (reference: Self et al. J Infect Dis. 2016; 213; 584-91. doi: 10.1093/infdis/jiv323)

RESPONSE: Added the following text along with the reference noted. 

Some respiratory viruses are frequently detected in healthy patients due to colonization, resulting in a questionable causative role in LRTIs.

page 2, line 49: I believe there are more than 90 pneumococcal serotypes

RESPONSE: Yes, this has been updated along with an appropriate reference.

page 2, line 53: should be Klebsiella pneumoniae

RESPONSE: Typo corrected.

page 2, line 54: would suggest adding a sentence related to etiology of CAP in children, where a pathogen was detected in 86% of 254 children with CAP in a relatively old but a well performed prospective study from Finland (Juven T et al. Pediatr Infect Dis J 2000; 19:293-8.

RESPONSE: Added the following text along with the reference noted. 

In a 3-year prospective study conducted in Finland, the etiology of CAP in children was detected in 86% of 254 children with 62% being viral, 53% bacterial and 30% coinfection. S. pneumoniae was the most commonly identified causative pathogen.

page 2, line 57: mortality should be changed to case-fatality rate

RESPONSE: Text updated.

page 2, Figure 1, left image: instead of Pseudomonas aeruginosa it should be aeruginosa

RESPONSE: Typo corrected in image along with updating the italics.

page 3, line 100: maybe include a sentence (and a reference) about temporal evolution of CRP during LRTI/pneumonia, i.e. it being low in the first 12-24 hours and reaching peak values on day 2-3 of illness.

RESPONSE: Added the following text along with the reference noted. 

PCT increases within 2-4 hours of infection, peaking at 24-48 hours.

page 4, line 136: there is another POCT using MxA and CRP that has been evaluated in patients with URTI and LRTI (Eur Clin Respir J. 2015; 2: 10.3402/ecrj.v2.28245.); moreover, RNA signatures have also been evaluated in children

RESPONSE: Added the following text along with the reference noted. 

A point-of-care test of CRP and Myxovirus resistance protein A (MxA) was used in 54 patients with pharyngitis or LRTIs to determine the etiology of the infection. This combination characterized 80% (16/20) with bacterial infection, 70% (7/10) with viral infection along with 92% (22/24) negative for a bacterial or viral infection. However, this study was small, and further confirmation of this point-of care test is needed. 

page 4, line 141: what is a relevant sample for determining etiology of LRTI? blood culture is relevant, BAL as well, tracheal aspirate is not always representative, sputum quality depends on patient effort and is useless in children. There have been studies looking for S. pneumoniae in blood with the use of PCR and it was superior to blood culture (Cvitkovič-Špik et al. Scand J Infect Dis 2013;45:731-7). I think these questions deserve some comments and discussion (with references) because they are clinically relevant.  There was a study using transthoracic aspirates that determined etiology of CAP in children in 47% of culture-negative patients (Vuori-Holopainen et al. Clin Infect Dis 2002; 34: 583-90.) Although the procedure seemed dangerous it was not associated with relevant complications.

RESPONSE: Added the following text along with the reference noted. 

In addition, the collection of optimal specimens to detect the pathogenic etiology of LRTIs must be considered, given the specifics of a testing modality as well as the logistics of obtaining the specimens. Specimens can be collected using invasive (blood, thoracentesis, transthoracic needle aspiration, bronchoscopic bronchoalveolar lavage or protected specimen brush) or non-invasive techniques (induced or expectorated sputum, nasopharyngeal swab, oropharyngeal/throat swab, and urine for antigen testing).  However, colonization of the respiratory tract with various pathogens must be taken into account when determining appropriate treatment regimens based on sputum or naso/oropharyngeal swabs. In a study of 340 patients with CAP, culture and RT-PCR were used to compare plasma and respiratory (nasopharyngeal swabs and sputum or tracheal aspirates) samples. In this study, RT-PCR for both plasma and respiratory samples identified S. pneumoniae more often than culture. However, good quality sputum samples are often challenging to obtain especially in the outpatient setting and from children.

page 5, Table: type of sample should be included with every test (in a separate column)

RESPONSE: Table 2 has been updated to include the sample types.

page 7, line 209: urine tests are not mentioned, although they are both reliable and fast, especially in adult patients with CAP: urine pneumococcal antigen and Legionella antigen – these tests require mentioning since they give reliable results rapidly and provide important information regarding selection of antibiotics

RESPONSE: Table 2 has been updated to include urine pneumococcal antigen and Legionella antigen tests. In addition, a paragraph in Section 4 has been added to describe these tests.

Reviewer 2 Report

Abstract:

L10:  Delete period between Streptococcus pheumoniae

Section 1.

L21-22:  Reference needed.

L23-24:  Reference needed.

L25-26:  Reference needed.

Section 2.

L36-37:  Reference needed.

L37-38:  Explanation of common bacterial coinfections would enhance the background information.

L39-40:  Reference needed.

L49-50:  Reference needed.

L60:  Where is VAP abbreviation/full spelling located previously?

L63:  Staphylococcus aureus was previously mentioned, so it can be abbreviated.

L65:  Capitalize "gram".

Figure 1:  Genus/species within the figure should be italicized.

L60-70:  Reference needed.

L73-74:  Reference needed.

Section 3.

L98-100:  Does this criteria only apply to LRTIs?

L109-110:  Are the decreases/reductions mentioned averages?  If so, it should be indicated.

L119:  Refer to the study name using parantheses.

L133-134:  Reference needed.

L135:  Space needed between "sodevelopment".

L135-136:  Reference needed.

L129-136:  Additional details/description would enhance this paragraph.

Section 4.

Table 2 - Is there a reason that some of the diagnostics listed are italicized and others are not?

L146:  Change "takes" to "take".

L158-159:  Are the "over 90% of pathogens" those that are all responsible for LRTIs?

L170:  What type of PCR?  Standard? Quantitative?

L173:  Times associated with PCR should be included in Table 2.

L184 and 189:  Should "panel" be capitalized?  Not consistent.

L193:  Are the semi-quantitative panels sPCR or qPCR?

Section 5.

L219-220: Specific examples would enhance the section.

L223:  Change "was" to "were".

L229:  Was C. difficile previouly mentioned?  If not, it would be helpful to include it within the narrative if it is going to be used as part of the conclusion.

Author Response

REVIEWER 2

Abstract:

L10:  Delete period between Streptococcus pheumoniae

RESPONSE: Typo corrected.

Section 1.

L21-22:  Reference needed.

RESPONSE: Reference added. 

L23-24:  Reference needed.

RESPONSE: Reference added. 

L25-26:  Reference needed.

RESPONSE: Reference added. 

Section 2.

L36-37:  Reference needed.

RESPONSE: Reference added. 

L37-38:  Explanation of common bacterial coinfections would enhance the background information.

RESPONSE: Added the following text along with the reference noted. 

Viral etiologies are now identified in 25% of patients, with many of these patients having bacterial coinfections with viral respiratory pathogens resulting in decreased mucociliary clearance of bacteria.  Examples of this pathology include influenza decreasing clearance of S. pneumoniae, and respiratory syncytial virus (RSV) inducing the adherence of S. pneumoniae, Pseudomonas aeruginosa, and Haemophilus influenza to airway epithelial cells.

L39-40:  Reference needed.

RESPONSE: Sentence deleted. 

L49-50:  Reference needed.

RESPONSE: Reference added. 

L60:  Where is VAP abbreviation/full spelling located previously?

RESPONSE: VAP has been defined in the text now. 

L63:  Staphylococcus aureus was previously mentioned, so it can be abbreviated.

RESPONSE: Abbreviation updated throughout text. 

L65:  Capitalize "gram".

RESPONSE: Typo corrected. 

Figure 1:  Genus/species within the figure should be italicized.

RESPONSE: Figure 1 has been updated. 

L60-70:  Reference needed.

RESPONSE: Reference added. 

L73-74:  Reference needed.

RESPONSE: Reference added. 

Section 3.

L98-100:  Does this criteria only apply to LRTIs?

RESPONSE: The references noted discuss the use of these CRP criteria for LRTIs.  Therefore, text was not added or changed.

L109-110:  Are the decreases/reductions mentioned averages?  If so, it should be indicated.

RESPONSE: The change in antibiotic consumption is a mean and so this has been added to the text. 

L119:  Refer to the study name using parantheses.

RESPONSE: Parentheses changed to include study name and abbreviation. 

L133-134:  Reference needed.

RESPONSE: Reference added. 

L135:  Space needed between "sodevelopment".

RESPONSE: Typo corrected. 

L135-136:  Reference needed.

RESPONSE: Reference added. 

L129-136:  Additional details/description would enhance this paragraph.

RESPONSE: Further details have been added regarding additional point of care combinations of tests. 

Section 4.

Table 2 - Is there a reason that some of the diagnostics listed are italicized and others are not?

RESPONSE: Table 2 has been updated; italicized diagnostics are categories, not the specific tests. 

L146:  Change "takes" to "take".

RESPONSE: Typo corrected. 

L158-159:  Are the "over 90% of pathogens" those that are all responsible for LRTIs?

RESPONSE: Text updated to further describe this, and reference included. 

L170:  What type of PCR?  Standard? Quantitative?

RESPONSE: ‘Standard’ has been added to the text to clarify. 

L173:  Times associated with PCR should be included in Table 2.

RESPONSE: Table 2 has been updated. 

L184 and 189:  Should "panel" be capitalized?  Not consistent.

RESPONSE: Panel has been capitalized for consistency. 

L193:  Are the semi-quantitative panels sPCR or qPCR?

RESPONSE: These are semi-quantitative real-time PCR (qPCR), and therefore additional text has been added as follows.  Note that semi-quantitative indicates above/below a certain amount of DNA molecules can be assessed. Reference has also been added.

Some of these approved panels provide semi-quantitative results, such as the BIOFIRE FilmArray System Pneumonia Panel plus which reports levels of organism concentration in genome copies/mL for values above 103.5 copies/mL for 15 bacterial pathogens.

Section 5.

L219-220: Specific examples would enhance the section.

RESPONSE: The text of that sentence has been updated to the following:

Due to the improper treatment of LRTIs, some infected patients may not be treated adequately because the responsible bacterium (such as S. pneumoniae, methicillin-resistant S. aureus and Gram-negative bacilli) is resistant to available antibiotics, leaving physicians without a weapon to combat the illness.

L223:  Change "was" to "were".

RESPONSE: Typo corrected. 

L229:  Was C. difficile previously mentioned?  If not, it would be helpful to include it within the narrative if it is going to be used as part of the conclusion.

RESPONSE: C. difficile has been mentioned in Section 3 as well in the following sentence. 

With the potential for antibiotic-resistant bacteria as well as sequelae from the use of unnecessary antibiotics including Clostriodes difficile infection, defining the etiology of the LRTI is imperative for appropriate patient treatment.